# Acoustic Performance of Sound Absorbing Materials Produced from Wool of Local Mountain Sheep

**DOI:** 10.3390/ma15093139

**Published:** 2022-04-26

**Authors:** Katarzyna Kobiela-Mendrek, Marcin Bączek, Jan Broda, Monika Rom, Ingvild Espelien, Ingun Klepp

**Affiliations:** 1Faculty of Materials, Civil and Environmental Engineering, University of Bielsko-Biala, Willowa 2, 43-309 Bielsko-Biala, Poland; kmendrek@ath.bielsko.pl (K.K.-M.); mbaczek@ath.bielsko.pl (M.B.); mrom@ath.bielsko.pl (M.R.); 2Selbu Spinneri, Amundalsvegen 91, 7540 Klæbu, Norway; ingvild.espelien@selbuspinneri.no; 3Consumption Research Norway SIFO, Oslo Metropolitan University, Stensberggata 27, 0130 Oslo, Norway; ingunk@oslomet.no

**Keywords:** wool, textiles, acoustic insulation, sound absorption, noise reduction

## Abstract

Wool of mountain sheep, treated nowadays as a waste or troublesome byproduct of sheep husbandry, was used for the production of sound-absorbing materials. Felts of two different thicknesses were produced from loose fibres. Additionally, two types of yarn, ring spun and core rug, were obtained. The yarns were used for the production of tufted fabric with cut and loop piles. During the examinations, basic parameters of the obtained materials were determined. Then, according to standard procedure with the use of impedance tube, the sound absorption coefficient was measured, and the noise reduction coefficient (NRC) was calculated. It was revealed that felt produced from coarse wool exhibits high porosity, and its sound absorbing capacity is strongly related to the felt thickness. For thicker felt the NRC achieved 0.4, which is comparable with the NRC of commercial ceiling tiles. It was shown that the crucial parameter influencing the sound absorption of the tufted fabrics was the pile height. For both types of yarns, when the height of the pile was increased from 12 to 16 mm, the NRC increased from 0.4 to 0.42. The manufactured materials made from local wool possess good absorption capacity, similar to commercial products usually made from more expensive wool types. The materials look nice and can be used for noise reduction as inner acoustic screens, panels, or carpets.

## 1. Introduction

For centuries, sheep wool was one of the important natural fibres commonly used for the production of apparel textiles, carpets, and blankets [1,2]. In recent decades, due to the rapid development of synthetic fibres and availability of competitive products in the market, the demand for wool has decreased significantly. In this situation, some new products designed mainly for technical applications were developed [3,4]. These new products include insulating materials [5,6,7], geotextiles [8,9,10], oil sorbents [11,12], and heavy metal absorbing materials [13,14,15].

The most applicable new wool products include insulating materials used in the construction industry. They provide good thermal and acoustic insulation and can be used as a substitute for common insulating materials such as mineral wool, polyester, and polyurethane foam [16,17]. Numerous investigations have revealed that materials manufactured from wool sufficiently dampen normal speech and possess excellent sound-absorbing ability, especially in regard to medium and high frequencies [18,19,20,21,22]. Contrary to other raw materials used for the production of soundproofing materials, wool is renewable, biodegradable, and naturally fire resistant. The enzymes break down the wool keratin into low molecular compounds, so wool buried in soil can degrade completely within several months [23,24]. Due to its high ignition temperature (close to 600 °C) wool is difficult to ignite [25]. When burning, it reveals a low propensity for flame spread, produces little smoke, does not melt or drip, emits low amount of toxic substances, and is easily extinguished.

Wool’s acoustic insulation properties were used to develop different commercial products. One group of products available on the market are mats, batts, or panels intended for insulation of roofs, walls, and ceilings [26,27,28,29,30,31,32]. They are designed mostly to insulate closed spaces and, as with other insulating materials, they need to be protected with covering materials.

The second group includes interior products, which are used to reduce noise inside houses, offices, and public facilities. These materials provide comfortable acoustic conditions and contribute to creating a pleasant indoor environment. The products are multifunctional, modify the microclimate, and not only improve the hearing experience but also influence other senses, such as sight, smell, and touch. They are designed to be exposed and decorate the interior. Due to their high moisture absorption ability and high heat of adsorption, wool materials prevent water condensation, regulate indoor humidity and temperature, and create a comfortable microclimate. Thanks to their ability to absorb volatile organic compounds, these products trap air pollutants and eliminate unpleasant smells in the room [33,34]. Even when heated, the products do not release the absorbed gases and may continue to purify the indoor air for up to 30 years.

Wool acoustic insulation materials used in interiors include screens/dividers, wall/ceiling panels, and rugs/carpets. Screens and space dividers are often used in offices to improve the acoustics of the workplace and to create private workspaces for employees in open office arrangements. Wall and ceiling panels installed either as single larger elements or as smaller tiles put together in clusters become a form of decoration and improve both the acoustic comfort and aesthetic qualities of the space. Rugs and carpets used as floor covers effectively reduce noise and minimize reverberation times.

The abovementioned products can be made using different techniques. The possible manufacturing methods include the already known traditional wool processing techniques: spinning, knitting, weaving, and tufting. Additionally, certain products can be manufactured using nonwoven and other unique techniques designed for special purposes.

Depending on the technique used, products with different structures and various acoustic performance can be developed [35,36,37,38]. The results of numerous studies of wool fabrics, nonwovens, and carpets can be found in the literature [39,40,41].

The review of the commercial products shows that sound absorbing materials offered on the market are manufactured from fine merino wool or other kinds of thin and soft wool, which is often transported from the other end of the world [42,43]. One can suppose that merino wool is more commonly used due to its greater availability on the world markets, rather than its superior sound absorption properties. Few attempts have shown that similar good sound absorption may be achieved using coarser wool obtained from local sheep breeds [44].

At the moment, local wool is discarded in many European countries and treated as an embarrassing and unnecessary byproduct of sheep husbandry [45,46,47,48,49]. Production of sound absorbing materials from this wool seems to be a reasonable alternative for its utilization.

During the investigations reported herein, several sound-absorbing materials designed for indoor applications were produced. The material used was a coarse sheep wool from the Polish local mountain breed. The insulating materials were manufactured by means of felting and tufting techniques. The acoustic performance of the materials, in dependence on their structural parameters, was analysed, and the sound absorption abilities of various products were compared.

## 2. Materials and Methods

### 2.1. Materials

#### 2.1.1. Raw Wool

The raw material used for the production of sound absorbing materials was wool of Polish Mountain Sheep. The breed is well adapted to long treks, steep and hardly-accessible pastures, and harsh climatic mountainous conditions. The fleece is mainly white with occasional brown or black variations (Figure 1). The wool used in the investigations was from the coloured variety.

The fleece of mountain sheep is not homogeneous. It consists of more delicate wool, which forms the undercoat, guarded hairs forming the outer coat, and kemp (Figure 2). The average diameter and length of wool equalled 36 µm with a standard deviation of 9 µm and 60 mm with a standard deviation of 16 mm, respectively. The same parameters for guarded hairs amounted to 64 µm, with a standard deviation of 11 µm and 120 mm with a standard deviation of 26 mm. The content of particular fractions varied considerably between individual animals. The wool content varied from less than 35% to as much as 65% of the fleece weight. The kemp content in the fleece usually equalled 5–20%.

#### 2.1.2. Preparation of Sound Absorbing Materials

The sound absorbing materials with different structures were manufactured according to the scheme (Figure 3). Felt and the yarns were manufactured in a spinning mill Selbu Spinneri (Klæbu, Norway). The mill is used for small scale wool processing and is equipped with carding and spinning machines supplied by Belfast Mini Mills (Belfast, PE, Canada) and Ramella (Biella, Italy). Tufted fabrics were manufactured in Sztuka Beskidzka (Czechowice-Dziedzice, Poland).

Felt was produced from the wool obtained from various fleeces without quality sorting. After scouring and removing plant contaminants, the raw wool was carded. During the process, an essential amount of kemp was removed. The machine used was the Felt Maker plate (Belfast Mini Mils). A layer of carded web was laid out on the bottom plate of the device and covered with transversely arranged layers. The four-layer stack was wetted with hot water and then with a solution of basic soap with pH 8. In a further step, the stack was loaded with the upper plate and felted for half an hour. Finally, the felt was washed and rinsed in warm water and dried at room temperature.

To produce the ring-spun yarns, the card web was formed into slivers, which were then drawn twice on a pin drafter. Drawn slivers were spun into yarns by the classic ring spinning technique. In this way, a single yarn 400 tex Z170 was obtained. In the next step, three single yarns were twisted together and finished. The final product was a three-ply yarn 400 tex Z 170 × 3 S 60; R 1240 tex (Figure 4a).

The core rug yarns were manufactured using the Rug Yarn Maker (Belfast Mini Mills). In the process, the card web was removed directly from the doffer of the carding machine and wrapped around a core formed from the previously spun wool yarn. The final product was a thick yarn with a linear density of 4 ktex and a diameter of 8–20 mm (Figure 4b).

The yarns were used to produce tufted fabrics with a manual pneumatic gun. The yarns were punched by the tubular needle of the gun into a prestretched primary backing mounted on a frame. The primary backing was a polyester plain weave cloth with the square set equal to 70 threads/dm and with both yarns’ linear density of 180 tex. To prevent the pile from being pulled out, a secondary backing was bonded with latex. The secondary backing was a leno fabric of polypropylene warp with a linear density of 33 tex and an acryl weft with a linear density of 125 tex. The warp and the weft sets for this fabric were 64 threads/dm and 40 threads/dm, respectively. Fabrics with cut and loop piles with a nominal height of 12 and 16 mm were obtained using the ring spun yarn (Figure 5). The core rug yarn was used to produce fabrics with loop piles of two different heights—12 and 16 mm (Figure 6).

### 2.2. Methods

#### 2.2.1. Measurement of Materials Parameters

The fibre thickness was determined with a light microscope coupled with the image Aphelion Dev software. The length of the fibres was determined separately for the guarded hair and the wool, in accordance with PN-ISO 6989:2000. The fibre surface morphology was examined by means of a JEOL JSM 5500 LV scanning electron microscope. The microscopic observations were carried out for fibres sputtered with gold in a JEOL JFC 1200 ionic sputter.

The surface density, thickness, and air permeability of the felt and tufted fabrics were determined in accordance with PN-EN ISO 9863–1:2007, PN-EN ISO 9864:2007, and PN-EN ISO 9237:1998 standards. The measurements were performed using a thickness gauge Tilmet 73 (Lodz, Poland) and a permeabilimeter FX 3300 Textest AG Instrument (Schwerzenbach, Switzerland). For each parameter, the mean value from 10 measurements and the 95% confidence interval was determined.

#### 2.2.2. Measurement of Sound Absorbing Coefficient

The acoustic performance of the materials was measured using a Brüel & Kjaer standing wave tube coupled with a loudspeaker placed at the end of it. During the measurements, a one-dimensional acoustic field was generated inside the tube. A plane sound wave was propagated in the tube and then reflected at the other end of it. As a result of the interference of the reflected wave with the incident wave inside the tube, a standing wave was formed with the sound pressure maxima and minima. By introducing the investigated sample in front of the hard termination, part of the energy of the incident wave was absorbed, and the energy of the reflected wave was reduced. As a consequence, the sound pressure maxima decreased and the minima increased. The sound absorption coefficient was calculated based on the ratio of the measured maximum and minimum standing wave sound pressure.

The measurements were carried out for round samples with the diameter of 100 mm in third-octave intervals within the frequency band of 250–3150 Hz. During the measurements, the sound absorption coefficient α was determined; then, the noise reduction coefficient (NRC) was calculated. The NRC was determined as the average value of the sound absorption coefficients at frequencies of 250, 500, 1000, and 2000 Hz.

## 3. Results

### 3.1. Properties of Materials

The values of the parameters of the sound absorbing materials are summarised in Table 1.

During the investigations, two felts with two different thicknesses—9.9 and 19.5 mm were obtained. For the thicker felt, which was twice as high as the thinner one, the surface density was respectively two times higher, while the air permeability was almost two times lower. The tufted fabrics were produced at two different pile heights—12 and 16 mm—set on the tufting gun. The measured thickness takes into account the pile layer, the backing fabrics, and the latex layer. With the additional layers, the thickness of the fabrics with the 12 mm piles equalled approximately 14 mm. For the 16 mm piles, the thickness reached approximately 16 mm for the ring spun yarn and even 18 mm for the core rug yarn. Consequently, the higher the pile, the higher the surface density. Compared to the fabrics made from the ring spun yarn, the surface density for the thick core rug yarn was greater. The increase in fabric thickness and surface density was strictly related to the decrease in their air permeability. For the ring spun yarn, when the height of the cut piles grew by only 2 mm, the air permeability dropped by 40%. The air permeability of the loop piles dropped even threefold.

Because of the too high fabric thickness, the air permeability measurement of the thicker fabric made from the core rug yarn was unworkable. In this case, the comparison between thinner and thicker samples was impossible.

### 3.2. Sound Absorption Capacity

The dependence between the noise absorption coefficient and the sound wave frequency for two samples of felt of different thicknesses is presented in Figure 7.

For both samples, at low sound wave frequency the α coefficient was low, from 0.1 to 0.15. Above 500 Hz, when the frequency grew, the α coefficient increased significantly, and at 2000 Hz, it reached a value of 0.79 for the thinner sample and 0.84 for the thicker one. At a frequency above 2000 Hz, a further increase in the coefficient was less meaningful. At the highest frequency of 3150 Hz, the α achieved the highest value of 0.89 and 0.95 for thin and thick felt, respectively. At all frequencies, the measured coefficient of the thicker sample was 0.15–2.2 higher.

Similar to the felt, the lowest value of the α coefficient for the tufted fabrics was observed at low frequencies. As the frequency increased, the α coefficient rose significantly until the frequency was 2000 Hz. Above this value, no further increase in the coefficient was observed (Figure 8 and Figure 9).

For the fabrics manufactured from the ring spun yarn, as the frequency increased from low to medium, the α coefficient increased from 0.15 to almost 0.9. For the fabric with cut piles, at the highest measured frequencies the coefficient achieved the value of 0.96 for the 12 mm piles and even 0.99 for the 16 mm piles. Similar maximum values were observed for the fabrics with loop piles. For this type of fabric, at all frequencies the coefficient was slightly higher in the samples with the higher piles.

The increments of the α coefficient were very similar for all the fibres manufactured from the core yarn. The coefficient grew steadily until 2000 Hz. Then, at higher frequencies, it achieved the maximum value of 0.96 for the 12 mm piles and 0.98 for the 16 mm piles.

The values of the Noise Reduction Coefficient (NRC) calculated for all the investigated samples are presented in Figure 10. The NRC was relatively low for the thinner felt. It was much higher for the thicker felt and reached the level determined for tufted fabrics. For all the tufted fabrics made both of the ring spun and the core rug yarns, the NRC value was between 0.4 and 0.43. For both types of yarn, the greater the pile height, the slightly higher the NRC value.

## 4. Discussion

The sound absorption ability of materials produced from mountain sheep wool is comparable with other wool acoustic insulation materials. Like other wool absorbers, the obtained materials possess low sound absorption capacity at low sound wave frequency and good capacity at medium and higher frequencies. The values of the sound absorption coefficient and noise reduction coefficient correspond well to those of commercial products made from merino wool and other wool articles produced with various textile techniques [50,51]. It is well known that sound absorption of insulating materials depends on many factors. For fibrous insulating materials consisting of randomly oriented loose fibres, it depends mainly on the fibre packing density and the product’s geometrical dimensions. For insulating textiles, the sound absorbing capacity is also strongly affected by the textile structure, which is the result of manufacturing technique and production parameters.

During the investigations, two techniques were applied to produce the sound absorbing materials: felting and tufting. Felting involves compression and consolidation of loose fibres by mechanical and chemical action in higher temperature and in a moist environment. The final product is felt which is likely the oldest textile structure known to mankind (Figure 11a). The felting ability is unique for the wool fibres, mainly thanks to the scales on the surface of the fibres (Figure 11b). During the reciprocal movements, the scales covering the adjacent fibres interlock with one another and form a compact felt structure.

Due to their low diameter, high crimp, and high scale frequency, fine wool fibres show good feltability [52,53]. Less flexible coarse fibres typical for mountain sheep are less prone to felting. Coarser fibres give felt a hairy appearance and significantly reduce its softness. Moreover, coarse fibres are usually accompanied by a higher proportion of stiff and brittle kemp, which considerably reduces wool feltability and even makes felting impossible. To improve the feltability of coarse wool, at least partial removal of kemp is necessary [54]. In the process of manufacturing felt from the mountain sheep wool, the kemp was partially removed during carding. The fibrous web obtained had sufficient felting ability and felt with reasonable quality could be formed.

Felt is composed entirely of tangled and physically interlocked fibres. They form a highly porous structure with a great number of built-in capillary channels and numerous interconnected pores in the micron and submicron scale. The porous structure provides high air permeability and accessibility of sound waves. Incidental sound waves entering the felt cause vibrations of air molecules accumulated within the pores between the fibres. The oscillating air molecules rub irregular interstices and lose their energy as a result of friction. Eventually, the energy of the sound wave is transformed into thermal and viscous heat, which is finally dissipated [26].

Intensive investigations into the sound absorption of the wool felt confirmed that it was a strongly dissipative and dispersive material with a strong damping effect [55,56]. High sound absorption capacity and extraordinary acoustic properties of wool felt have been long known and are commercially utilized in piano manufacturing. For almost two centuries, it has been used to coat piano hammers, to produce piano dampers, as well as isolating pads ensuring vibration isolation between the vibrating strings and the cast iron frame.

In housing construction, wool felt is commercially applied for the production of interior soundproofing materials to improve the acoustic comfort and reduce the reverberation resonances in the room [36]. For these materials the NRC, which is a quantitative measure of a material’s potential for reverberation control, is above 0.4.

In the investigated samples, the NRC of the thinner felt equals 0.31, which was approximately 0.1 below the level of commercial products. The NRC of the thicker felt was higher, and its values were comparable with those of the commercial soundproofing ceiling tiles.

Similarly to fibrous insulating materials formed from randomly oriented fibres, the acoustic properties of felt depend on its packing density and thickness. The packing density of felt made from coarse, mountain sheep wool is relatively low. In this case, empty spaces between the fibres were larger, and the porosity of the felt was higher in comparison to the felt obtained from fine fibres [57]. The felt had low surface density and high air permeability. As a result, the NRC of the thinner sample was lower. The air permeability decreases with greater thickness, and the path length of the sound wave passing through the material is longer. Thus, viscosity losses, which convert acoustic energy into heat, are greater, and sound absorption is more effective [21,30].

The second technique used for the production of insulating materials involved stitching yarn into the backing fabric to create loop, cut, or mixed pile with a different height. Wool tufted carpets are highly effective in controlling noise in buildings. The carpets absorb airborne sound, reduce surface noise generation, and the impact of sound transmission between the stories in multi-storey buildings. Additionally, tufted carpets virtually eliminate floor impact sounds such as the noise produced by footfalls, furniture moved across the floor, and objects dropped on the floor.

During the investigations, the tufting technique was used to produce tufted fabrics from ring spun and core rug yarns.

The ring spun yarn manufactured with the classic ring spinning technique was composed of twisted fibres laying around one another in concentric helical paths. Friction forces between the fibres enhanced the adhesive forces and prevented the fibres from slipping under the tensile strain. The yarn had properties that enabled the production of fabrics with the tufting technique. It was flexible enough to provide loop piles and had a relatively low thickness to be easily cut with a knife of a tufting gun. Tufted fabrics with both cut and loop piles with two different heights were easily obtained from this yarn. For both cut and loop piles, the determined coefficients achieved the same value at the same pile height. For both types of piles, higher sound absorption was observed for the higher piles.

The other type of yarn, namely the core rug yarn, was obtained by twisting a fibrous web around the core yarn produced in a separate process. The structure of the core rug yarn was less compact, which creates the risk of the fibres being pulled out during pile cutting. Due to this risk and the high linear density, which hindered cutting and causes fast blunting of the tufting gun knives, it was impossible to form fabric with cut piles. For this reason, for core rug yarn only fabrics with loop piles were obtained. The sound absorption coefficients for these fabrics were similar to the ones for the fabrics obtained from the ring spun yarn, and as with the ring spun yarn, the NRC was higher at greater pile height.

Obviously, increasing the pile height for both types of yarns was connected with a greater thickness and a higher surface density of the tufted fabrics. This change led to a significant reduction in air permeability, which greatly affected the sound absorption capacity. Moreover, higher piles are more prone to deformation, which in turn increases their tortuosity and lengthens the path of the sound wave moving inside the fabrics.

The results obtained show that the acoustic performance of the tufted fabrics was not affected by the structure and parameters of the pile yarns. The results confirmed the previous findings of other authors, namely that the thickness of the fibres and structure of the pile yarn have relatively less effect on the acoustic performance of tufted fabrics [58]. The investigations clearly show that the noise absorption coefficients were strongly dependent on the pile height. This finding is consistent with previous results obtained for wool, acrylic, and polypropylene carpets [59,60,61].

The sound absorbing materials were manufactured from coarse wool of naturally coloured fleeces. For years, white fibres were more appreciated and had greater economic value because they could be dyed different colours. The investigation showed that the coloured variety can serve as valuable raw material to manufacture products with good acoustic performance. In this way, ecologically correct, environmentally sustainable, attractive, more rustic, and natural-looking materials can be obtained. Using a combination of white and naturally coloured wool allows obtaining various colouristic effects of the final products.

## 5. Conclusions

Coarse wool obtained from mountain sheep can be used to produce attractive sound absorbing materials. The sound absorption coefficients of these materials were similar to those of other products available on the market, usually made from more expensive wool types. The materials produced, both felt and tufted fabrics, can be successfully used for reverberation control and improvement of noise comfort in rooms. In addition to their insulating functions, the materials can also be decorative. The use of local wool to produce sound absorbing materials can serve as a reasonable solution for the utilization of wool obtained from the mountain sheep. In this way local wool treated in many European countries as waste can be used to produce valuable products.

Acoustic properties of the produced materials depend on their structure and parameters. The sound absorption of felt formed from randomly oriented, tangled, and physically interlocked fibres depends mainly on the fibre packing density and felt thickness. Wool from the mountain sheep is thick; therefore, it is difficult to achieve dense packing of these fibres. Less packed felt is highly porous and, consequently, possesses higher air permeability. As a result, its sound absorption is lower, compared to felt made from fine wool, and may be insufficient for some applications. A higher level of sound absorption can be achieved through increasing the felt thickness.

Tufted fabrics can be manufactured from yarns with different structure and parameters. The ring spun yarn produced from coarse wool has adequate flexibility to be used to produce fabrics with both cut and loop piles. The core rug yarns are much thicker, stiff, and difficult to cut with a tufting gun; so, only fabrics with loop piles can be made of this yarn. The influence of the structure, pile parameters, and the type of piles on the absorption capacity of tufted fabrics is less distinguished. The absorption capacity of fabrics manufactured from both ring spun and core rug yarns depends on the pile height.

## Figures and Tables

**Figure 1 materials-15-03139-f001:**
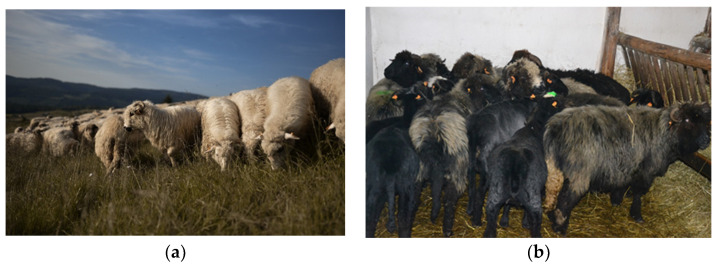
Polish Mountain Sheep; (**a**) typical white variety in the mountain pasture; (**b**) black variety during shearing.

**Figure 2 materials-15-03139-f002:**
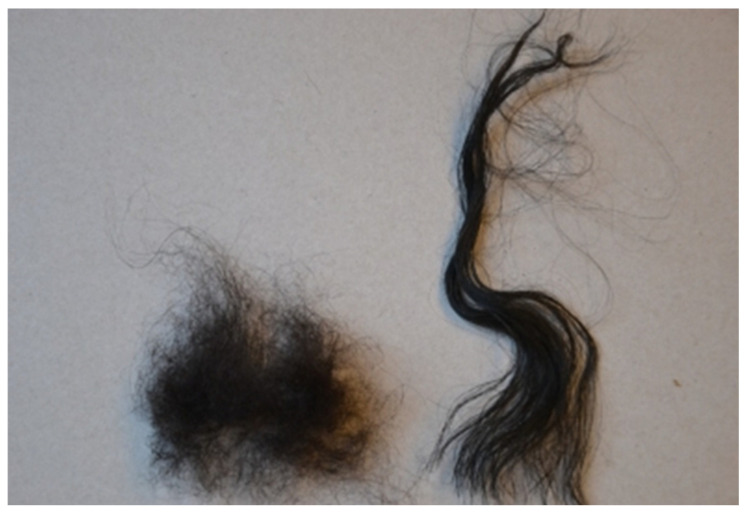
Wool and guarded hair forming the fleece of mountain sheep.

**Figure 3 materials-15-03139-f003:**
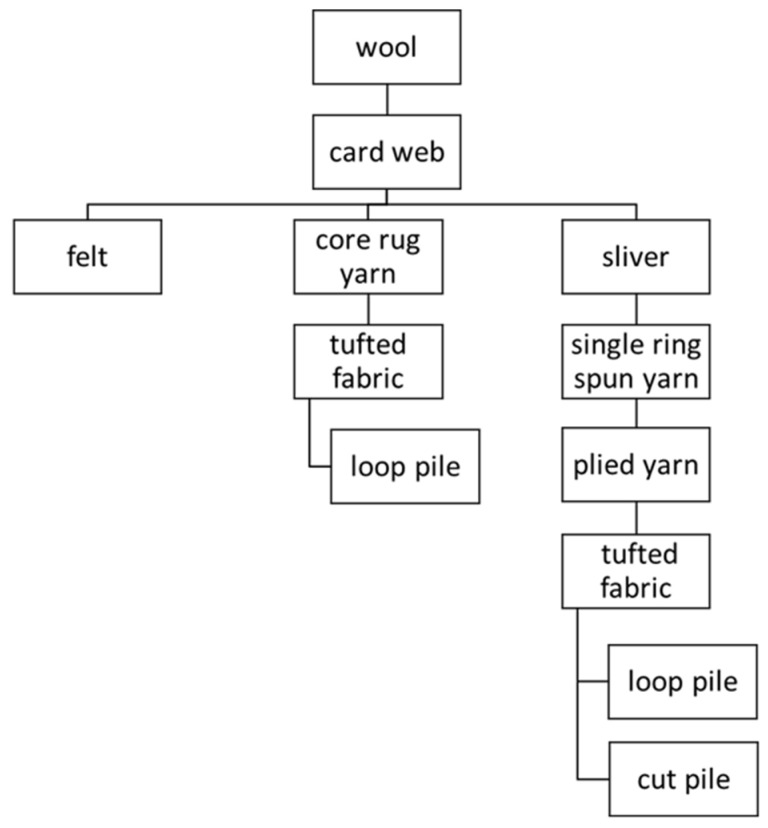
Insulation materials’ production scheme.

**Figure 4 materials-15-03139-f004:**
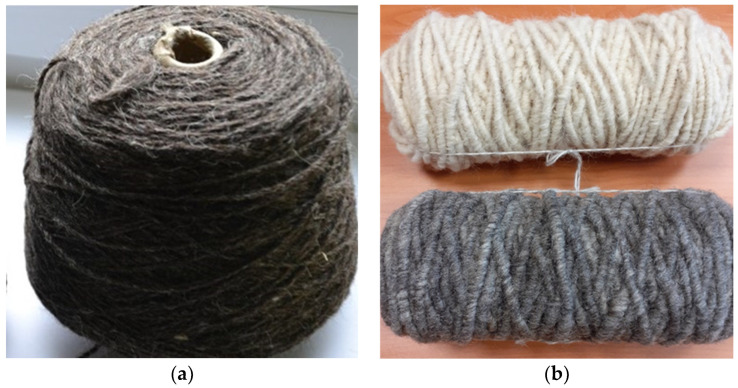
Wool yarns manufactured from mountain sheep wool; (**a**) ring spun; (**b**) core rug.

**Figure 5 materials-15-03139-f005:**
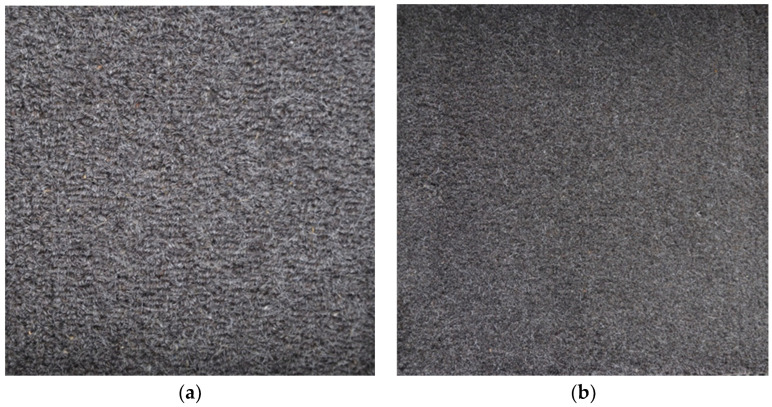
Tufted fabrics manufactured from ring spun yarn; (**a**) loop pile; (**b**) cut pile.

**Figure 6 materials-15-03139-f006:**
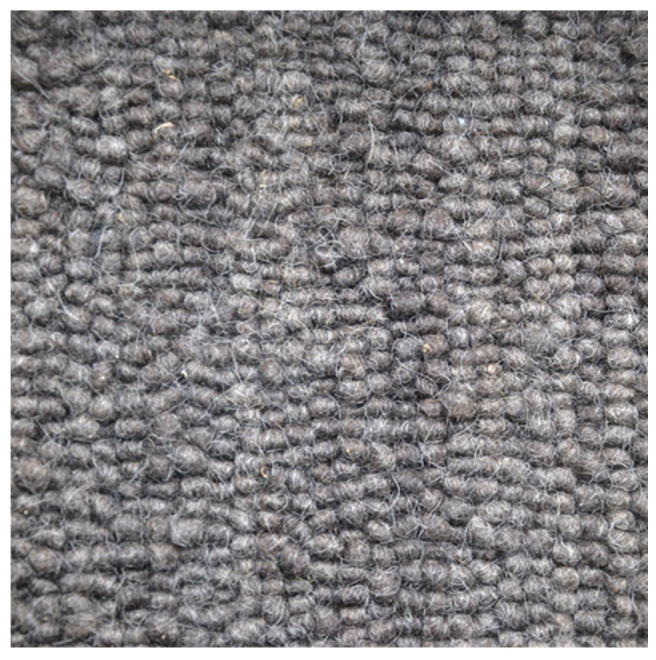
Tufted fabrics manufactured from core rug yarn.

**Figure 7 materials-15-03139-f007:**
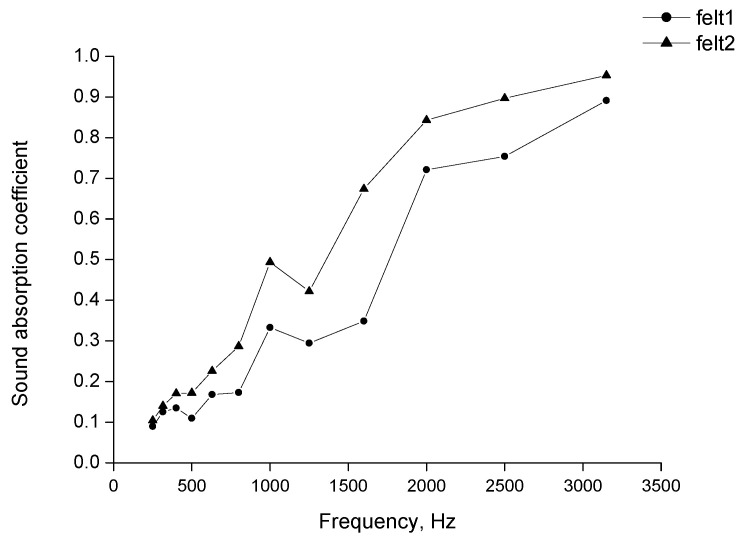
Sound absorption coefficient in normal incident for wool felts.

**Figure 8 materials-15-03139-f008:**
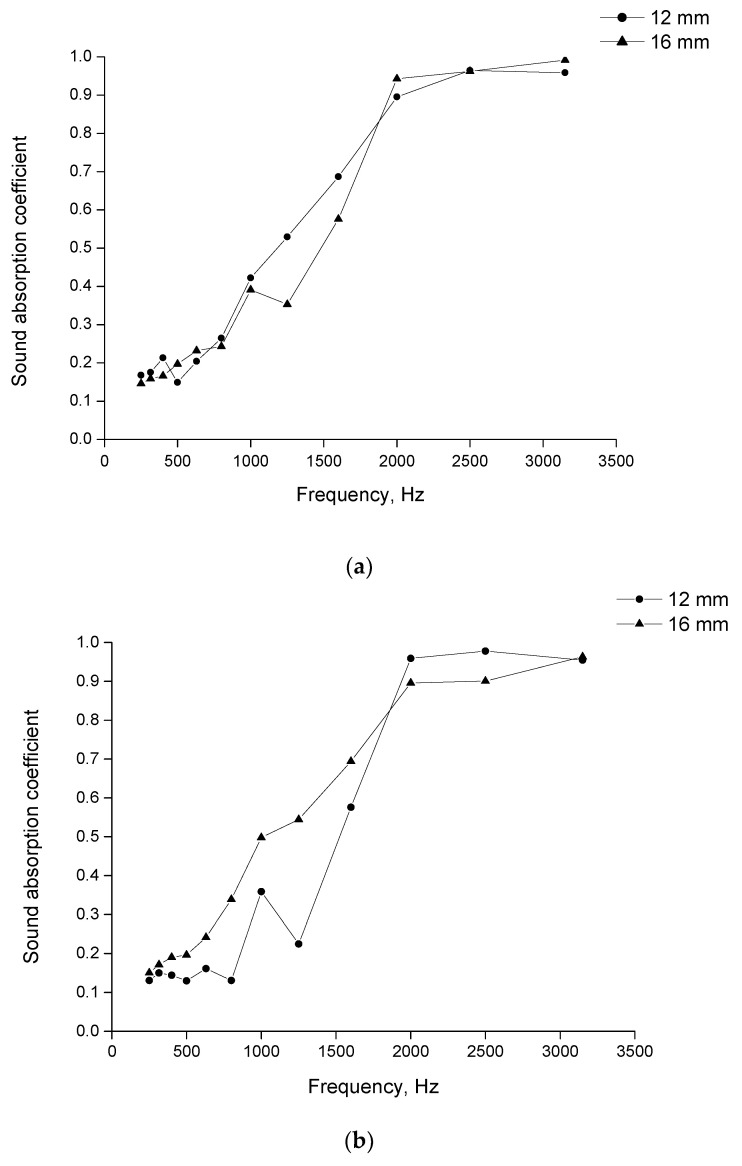
Sound absorption coefficient for tufted fabric made from ring spun yarn; (**a**) cut pile; (**b**) loop pile.

**Figure 9 materials-15-03139-f009:**
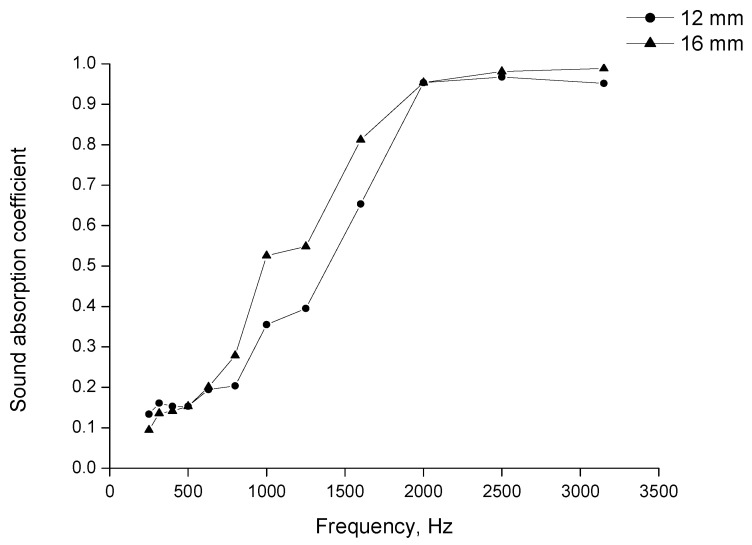
Sound absorption coefficient in normal incident for tufted fabric made from core yarn.

**Figure 10 materials-15-03139-f010:**
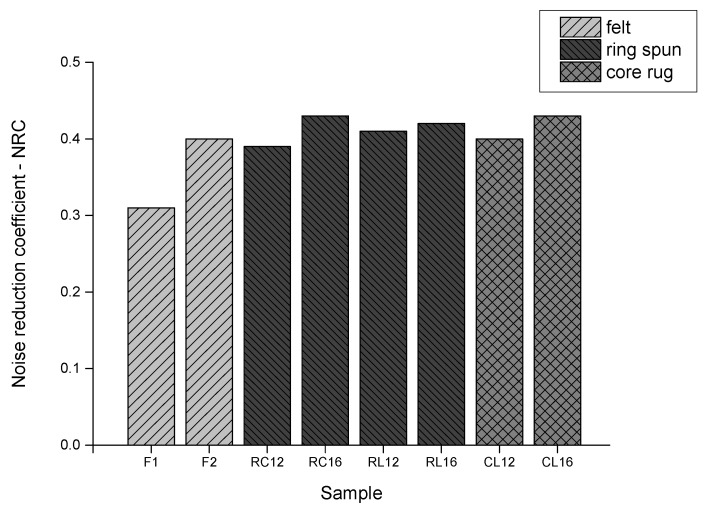
NRC value for felt and tufted fabrics.

**Figure 11 materials-15-03139-f011:**
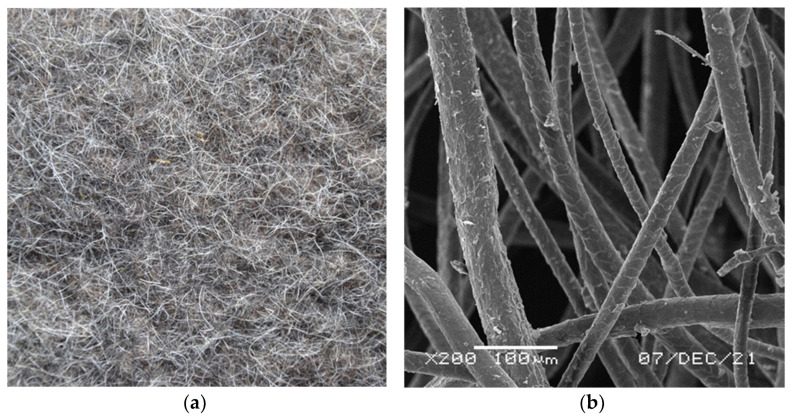
Wool felt; (**a**) tangled fibres on the surface; (**b**) fibres of different thickness covered with scales.

**Table 1 materials-15-03139-t001:** Basic parameters of sound absorbing materials produced from mountain sheep wool.

Material	Symbol	Thickness mm	Surface Density g/m^2^	Air Permeability l/m^2^/s
felt	F1	9.9	815	890
	F2	19.5	1550	512
ring spun yarn cut piles	RC12	14.0	3700	98
	RC16	16.0	4100	60
ring spun yarn loop piles	RL12	14.9	3500	230
	RL16	15.8	4200	80
core rug yarn	CL12	14.2	4000	173
	CL16	18.1	4800	-

## Data Availability

All data are presented in the paper.

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
