# Peer review of "Acoustic Performance of Sound Absorbing Materials Produced from Wool of Local Mountain Sheep"

_materials, 2022, doi:10.3390/ma15093139_

Round 1
Reviewer 1 Report
In the manuscript entitled "Acoustic performance of sound absorbing materials produced from wool of local mountain sheep", the authors show the potential of local mountain sheep wool to more commonly used merino sheep wool for preparation of sound absorbing materials. I have following comments, recommendations and questions:
- Materials and Methods section: I recommend authors move the parts between lines 127 and 197 to methods section and divide the methods sections into relevant subsections (e.g. methods of preparation of sound absorbing materials, measurement of thickness, acoustic performance etc.) to make the manuscript clearer and easier to follow for a reader.
- I wonder why the two felts of particular thicknesses were chosen.
- Figures: Resolution of all figures must be improved especially figures: 7,8, 9.
- My biggest concern with this study is the that whether the values obtained (noise reduction coefficient and sound reduction coefficient) for different samples are statistically different. Did authors do replications and run statistics?
- As the authors emphasize that local mountain sheep wool could be an alternative to merino sheep wool, the author should compare and discuss their result more clearly and succinctly with respect to similar sound absorbing materials made from merino sheep wool.
- Improvement in English in terms of grammar and style is necessary, in many places the sentences are not clear.
Author Response
1.Materials and Methods section: I recommend authors move the parts between lines 127 and 197 to methods section and divide the methods sections into relevant subsections (e.g. methods of preparation of sound absorbing materials, measurement of thickness, acoustic performance etc.) to make the manuscript clearer and easier to follow for a reader.
According to reviewer suggestion the section was divided into relevant subsections.
2.I wonder why the two felts of particular thicknesses were chosen.
The thinner felt was produced from a four-layer stack of fibrous web. For this felt the NRC was determined. The obtained value was approximately 25% lower than the NRC measured for tufted fabrics. To improve NRC the second sample of felt with bigger thickness was obtained. By increasing felt thickness sound absorption was improved and the level of NRC obtained for tufted fabrics was achieved.
3.Figures: Resolution of all figures must be improved especially figures: 7,8, 9.
According to reviewer comment the resolution of figures 7, 8 and 9 was improved.
4.My biggest concern with this study is the that whether the values obtained (noise reduction coefficient and sound reduction coefficient) for different samples are statistically different. Did authors do replications and run statistics?
The acoustic performance of the investigated materials was studied with the use of an impedance tube according to the procedure described in the methods section. During measurements, the sound absorption coefficient was determined according to a standard procedure in third-octave intervals within the frequency of sound wave of 250 – 3150 Hz. For comparison between samples, the NRC as the average value of the sound absorption coefficients at four frequencies 250, 500, 1000, and 2000 Hz was calculated. The measurements were performed for round samples with a diameter of 10 cm cut at random places of the materials. During measurements of surface density and thickness it was found that the combined standard uncertainty of the surface density does not exceed 10% of the mean value for each material and the combined uncertainty of the thickness is even smaller. These values allow to classify the textile materials as homogeneous.
The measurements of the sound absorption coefficient, like all other physical measurements, are burdened with measurement errors. Error sources include bias and random errors. The analysis of errors during measurement with impedance tube is a topic for a separate paper (for example: J. Niresha, S.Neelakrishnana, S.Subha Rani. Investigation and correction of error in impedance tube using intelligent techniques Journal of Applied Research and Technology 14 (2016) 405–414) and was not the goal of conducted investigations.
Taking into account the samples uniformity, the goal of the research and the measurement precision required to achieve this goal, the multiple measurements and statistical analyses were not performed.
- As the authors emphasize that local mountain sheep wool could be an alternative to merino sheep wool, the author should compare and discuss their result more clearly and succinctly with respect to similar sound absorbing materials made from merino sheep wool.
The authors do not emphasize that local wool can be an alternative to merino sheep wool. There are no such statements in the text. Local wool has quite different characteristics and for many applications local wool cannot replace merino wool.
As a result of performed investigations, it was stated that the coarse local wool from mountain sheep, can be used to produce attractive sound absorbing materials with acoustic characteristics comparable with products made from merino or other wool types. The investigations shown that the local wool, treated in many European countries as waste, can be used in reasonably way to produce valuable products. To our knowledge the paper is unique. There is no other paper on the acoustic properties of felt and tufted fabrics produced from local wool.
Generally, the scientific literature on the acoustic performance of felt and tufted fabrics made from different types of wool is scarce. The studies presented in the literature are concentrated on fibrous sound absorbing materials obtained using other techniques. Usually, the papers do not consider the wool origin and rarely take into account its characteristic. Additional information on the acoustic performance of wool products is presented in technical documentation of producers. The comparison made in the paper was based on this information. Nevertheless, the technical documentation is usually limited to the final NRC value and does not contain information on the type of wool used for the production and other details on product manufacturing. The more precise information is difficult to access. The statement that the commercial sound absorbing materials are produced from merino and other wool types is based on the market research performed by the team from Oslo university. This report is quoted in the paper.
Taking into account the above statements, the suggested by reviewer comparison and more clear and succinct discussion of the obtained results with results of other authors is impossible.
To avoid confusion in this matter, some parts of the text have been changed.
6.Improvement in English in terms of grammar and style is necessary, in many places the sentences are not clear.
According to reviewer comment the paper was revised and English was improved.
Reviewer 2 Report
The manuscript is well organized. However, there are minor things to add up in the revised version to promote the manuscript, more comprehensive and interesting.
The comments are as follows;
1) The abstract should be modified to be more informative and relevant to the main findings. It would be more informative if the results were expressed in numbers, i.e., quantitatively.
2) The author should summarize and discuss the dominant properties of the best material in the present work and such as sound absorption capacity, the noise reduction coefficient the advantages and disadvantages in the table, and compare it to that of the conventional materials reported in the literature.
This strategy could ease understanding for other researchers and convince them that this finding is promising material.
Author Response
1) The abstract should be modified to be more informative and relevant to the main findings. It would be more informative if the results were expressed in numbers, i.e., quantitatively.
According to reviewer suggestion, the abstract was modified. Relevant information was added and the main findings were underlined.
2) The author should summarize and discuss the dominant properties of the best material in the present work and such as sound absorption capacity, the noise reduction coefficient the advantages and disadvantages in the table, and compare it to that of the conventional materials reported in the literature. This strategy could ease understanding for other researchers and convince them that this finding is promising material.
The term “best material” is relative. For some applications, the best car may be the car with the lowest dimensions, while for others the big car with the powerful engine.
The goal of the research was not to find “best material”. The goal, which is defined at the end of Introduction, was to produce sound absorbing materials from the mountain sheep and compare their sound absorption abilities. This was done and discussed in the paper.
The comparison of the performance of obtained materials with the conventional materials reported in the literature is difficult. Firstly, the literature on wool absorbing materials produced by felting and tufting is scarce. Secondly, the available information on conventional sound absorbing materials does not take into account the origin of the wool, its characteristics and the method of material manufacturing. Comparison between materials made from wool of different origins and products made with different techniques has no sense.
Round 2
Reviewer 1 Report
The authors addressed most of the comments and concerns to satisfactory level.